# Effectiveness of carbon dioxide cryotherapy for the treatment of cutaneous leishmaniasis: Systematic review and meta-analysis

**Feleke Tilahun Zewdu** [1] *, **Bisrat Misganaw Geremew**[2], **Endalamaw Gadisa Belachew**[3], **Kassahun Alemu Gelay**[2]

1 Dermatology Department, Boru Meda General Hospital, Dessie, Amhara, Ethiopia, 2 Institute of public health, University of Gondar, Gondar, Amhara, Ethiopia, 3 Armauer Hansen Research Institute, Addis Ababa, Ethiopia

* momflk@gmail.com

## Abstract

### Background

Cutaneous leishmaniasis is one of the neglected tropical diseases which is hard to treat. Carbon dioxide-based cryotherapy is a novel therapeutic option for cutaneous leishmaniasis in both developed and developing nations. This study aims to summarize the pooled evidence on the effectiveness of carbon dioxide-based cryotherapy for the treatment of cutaneous leishmaniasis.

### Methods

Searches of grey literature using Google Scholar and databases including PubMed, Scopus, EMBASE, Web of Science, and Google Scholar were conducted to find studies that reported the cure rate of cryotherapy. The search, screening, data extraction, and critical evaluation were carried out by two authors, with a third acting as a tiebreaker. To locate papers, we used Medical Subject Headings (MeSH) phrases and keywords. Between May 10 and May 13, 2023, the review protocol was developed using the Preferred Reporting Items for Systematic Reviews and Meta-Analyses (PRISMA) 2020 checklist. For the meta-analysis, STATA 17 was the statistical software used. The random effects model was employed to compile the effect estimates. Lastly, we used the funnel plot and Egger's test to evaluate publication bias, heterogeneity, sensitivity, and subgroup analyses.

### Result

Out of the 16 researches included in the review, seven papers with a total of 1,357 cases were chosen for the meta-analysis. For the treatment of cutaneous leishmaniasis, the pooled cure rate with carbon dioxide-based cryotherapy was 87.84% (95% CI: 65.92–109.77). Randomized controlled trials made up the majority of the studies that were used. The highest cure rate was seen when there were two lesions, 94.34% (95% CI:68.21–119.48), and when the size of the lesion was less than or equal to 4, 93.83% (95% CI:68.92–118.75). There was no indication of a significant publishing bias.

**Data Availability Statement:** All the necessary data/files are within the Dryad database https://doi.org/10.5061/dryad.fttdz092v.

**Funding:** The author(s) received no specific funding for this work.

**Competing interests:** The authors have declared that no competing interests exist.

## Conclusion

Carbon dioxide-based cryotherapy revealed a high pooled efficacy. The efficacy of the CL instances was mostly dependent on the extent of the lesion and the frequency of therapy administration. To determine if this therapy is beneficial in a routine care context, a large-scale study with a sound design is necessary.

### Author summary

Neglected tropical diseases like cutaneous leishmaniasis are often overlooked. It can take on a variety of clinical forms and presentations. Comparably, the various causal agents and clinical forms of the treatment also vary. Both localized and systemic treatments are effective in treating localized cutaneous leishmaniasis. In most dermatological clinics, liquid nitrogen cryotherapy was the primary method employed. However, most dermatology clinics also utilize carbon dioxide-based cryotherapy sparingly, although the gynecology department uses it often for cervical cancer screening. On the other hand, the product is accessible, simple to use, painless, reasonably effective, and leaves fewer scars. Carbon dioxide cryotherapy had a pretty high pooled impact across trials, offering another accessible, easy-to-apply, and reasonably effective therapeutic option for localized cutaneous leishmaniasis.

## Introduction

The clinical variants of CL, their accessibility, and their complexity all influence how CL is treated [1,2]. Treatments called pentavalent antimonials are the cornerstone of CL therapy. On the other hand, systemic use of liposomal amphotericin B (AmBisome), sodium stibogluconate (SSG), meglumine antimoniate, paromomycin, and miltefosine. Pentavalent antimonial is given parenterally in many regions of the world; but, because of its toxicity, it cannot be used in areas with inadequate infrastructure. Therefore, baseline laboratory monitoring examinations are necessary. The most popular therapy, SSG, necessitates painful and dangerous intramuscular injections (IM) every day. Among the most severe adverse effects are renal toxicity, pancreatitis, and cardiotoxicity [3,4].

Localized cutaneous leishmaniasis can be treated with systemic anti-leishmanial drugs or by allowing the patient to recover on its own [5,6]. Additionally, unknown plants, mud, heat, and other remedies can be used by community traditional healers. Patients may then experience scarring following application, increasing their risk of stigma and other psychological risks [7,8]. Nonetheless, there are several approaches to treating LCL, such as topical medications, intralesional antimonial, thermotherapy, cryotherapy (using both liquid nitrogen and carbon dioxide base), and others. Although they don't require daily application or injection, they are comparatively non-toxic. However, if used deeply, there may be some injection site discomfort and scarring [3,9–11].

According to the WHO and the National CL treatment guideline states that uncomplicated LCL lesions should either not be treated at all or should only be dealt with by local agents. LCLs that are recommended to be treated with local agents are lesions with a size of the lesion less than 5cm, lesions less than 4, site of the lesion not at the cosmetic area, joint, and mucosa. Furthermore, when the treatment and prevention of these cases (LCL) is not done at the health institution, the CL patients will go to the local traditional healers. Application of unknown

traditional treatment with unknown strength will result in a scar, hyperpigmentation, superinfection, and severe pain at the application area. This traditional application of unknown agent-related effects results in stigma. Thus, the patient's acceptability and preference for modern CL treatment needs further research.

Furthermore, LCL can be treated with topical agents like cryotherapy, thermotherapy, topical creams (paromomycin), and IL SSG. Cryotherapy can be carbon dioxide and a liquid nitrogen base. However, these had different cure rates, adverse events, accessibility, and modes of delivery Carbon dioxide-based cryotherapy is available at health institutions including health centers to be applied by trained nurses, health officers, and general practitioners. It was primarily intended to be used for cervical cancer screening and treatment. Carbon dioxide can be found at the soft drink and beer factories. It is simple to use [12], effective with more than 90 cure rates [11,13], has relatively few adverse events [3,14], and the application lesion takes less time to heal [11,14]. However, liquid nitrogen-based cryotherapy is only available in dermatological clinic hospitals in Addis Ababa (ALERT), Gondar (Gondar University Comprehensive Specialty Hospital), and Bahirdar (Agriculture Institute) in the Amhara region. Nevertheless, the healing time is unpredictable, frequent applications are required, scar formation is severe, and the cost ranges from 60 to 300 ETB. Although this type of treatment is used in a few dermatological clinics and is included in the national treatment guideline [9], its efficacy has yet to be properly investigated in Ethiopia.

### The rationale of the review

The treatment of cutaneous leishmaniasis depends on the species of the causative agent, clinical forms of CL, clinicians' decisions, experience of the clinicians, and availability and cost of the treatment. Localized cutaneous leishmaniasis can be treated with local therapies including cryotherapy (Carbon dioxide and liquid nitrogen base), topical paromomycin, intralesional sodium stibogluconate, and others. However, most of the treatments of the CL caused by *L. aethiopica* are generally unsatisfactory [13].

The effectiveness, adverse events the duration of the treatment, and the schedule to be administered are different from country to country. In the Old World countries, the effectiveness of carbon dioxide-based cryotherapy ranges from 63.5% to 98.7% [15].

## Method and materials

### Protocol and registration

The Preferred Reporting Items for Systematic Reviews and Meta-Analyses (PRISMA) 2020 checklist was used to develop the review protocol [16]. This review has been registered at PROSPERO with registration identification number CRD42023408528.

### Eligibility criteria

Primary studies of all types of study designs, published in the English language, and one reported outcome were included in the review. Both published and unpublished articles with no limitation to the study period until May 2023 were included. However, articles without full text and abstracts, editorial reports, letters, reviews, and commentaries were excluded from the study.

### Search strategy

The database search was carried out between March 16–29, 2023, using electronic databases such as PubMed, Google Scholar, Global Health, Scopus, EMBASE, and African Journal

Online (AJOL) and grey literature (locally published papers, conference presentations, and dissertations). The last database search was done on March 29, 2023. We used MeSH headings and free text terms without restricting the study design or period (*S1 Table*).

## Study selection and data extraction

The Mendeley software, author names, location and setting, participant counts, study dates, and study duration were all used to remove duplicate papers found in various databases. The inclusion and exclusion criteria of the review were also used for the study selection process. To ensure the homogeneity of our search, each reviewer independently selected the appropriate papers for the review, which were then gathered. This method was in line with the Cochrane Review Handbook's fundamental guidelines for choosing studies and extracting data, which note that data may be presented in several formats but are commonly translated into a format appropriate for meta-analysis. Additionally, multiple reports of the same study need to be linked together, and data should be extracted from study reports by at least two people, independently. So that this review employed two individuals for both study selection and data extraction. Both experts received a set of agreed-upon inclusion and exclusion criteria. Each of them evaluated the articles and decided which ones to include or leave out of the analysis.

## Outcome measurement and quality assessment

Data extraction was made after a careful review of outcome measurement. This study aims to summarize the pooled evidence on the effectiveness of carbon dioxide-based cryotherapy for the treatment of cutaneous leishmaniasis.

Different studies have used Carbon dioxide cryotherapy in various ways. It was administered either once a week [17], every two weeks [14,18–22], every month [23–28] every 2 months [21] and some did not mention at all [29,30]. The course of treatment was followed for the entire term until the lesion healed [14,22,23,26,27], for three months [17,18,25,29] and not mentioned at all [19–21,24,28,31] at a total dose of four to six applications [17,20], three to five applications [21,30,31] and the rest did not mention the number of cryo- applications [14,18,19,24–26,29,32]. Moreover, some studies [21–23,26] also state the duration of cryotherapy application continued until the lesion healed with no set dosage limit.

Following the administration of carbon dioxide cryotherapy, cases of cutaneous leishmaniasis were evaluated using clinical parameters such as size reduction, flattening, re-epithelialization, and erythema resolution to ascertain whether or not the CL lesion healed [4,33,34].

Only a few publications describe the devices used to treat cutaneous leishmaniasis. The equipment used to treat cutaneous leishmaniasis includes the SubZero Cryotherapy machine [17,22,24], and the MedGyn 2000 [20], and some publications failed to describe the types of devices utilized [14,18,19,21,23,25–29,31].

Furthermore, to check the quality of the study article, the Newcastle—Ottawa quality assessment scale (adapted for observational studies: case-control and Observational-cohort study design) and risk of bias assessment for randomized studies (RoB 2) [35] were used. FT tried to grade the quality of the articles using an evaluation tool that has three parts: selection (5 stars), comparability (2 stars), and outcome (3 stars). Selection took into account the sample's representativeness, sample size, non-respondents, and determination of the exposure (risk factor), while comparability took into account whether or not the subjects in various outcome groups are comparable based on the study's design or analysis and factor-controlling. Additionally, the outcome took into account the evaluation of the result and statistical analysis of each of the included articles (**S2** and **S3** **Tables**).

## Data synthesis and statistical analysis

A Microsoft Excel spreadsheet was used to extract the data, which was then imported into STATA 17 for analysis [36]. Tables, figures, and forest plots were used to describe and summarize the major investigations. These studies included in the studies conducted throughout the world the effectiveness of carbon dioxide-based cryotherapy ranges from 64 to 100% of the cure rate. Therefore, to handle this variability between the studies, the random effects model is the best one. So a random-effects model with a 95% confidence interval was used to pool the cure rate of carbon dioxide base intervention for cutaneous leishmaniasis. Cochran's Q and $I^2$ statistics have been used to evaluate the heterogeneity between studies. The percentages of about 25% ($I^2 = 25$), 50% ($I^2 = 50$), and 75% ($I^2 = 75$) would, respectively, indicate no, moderate, medium, and high heterogeneity [37]. To identify the apples-and-oranges problem of a meta-analysis, subgroup analysis, meta-regression, and the Galbraith plot test were conducted. Egger regression tests and visual assessment of funnel plot asymmetry were used to determine 'the file drawer' problem of meta-analysis (publication bias).

Furthermore, all these data including the data extraction table, analysis output, and others are within the Dryad database [15].

## Result

### Search results

A total of 1,390 articles were included. Among these 1,141 articles were removed due to duplication and 212 were removed after reviewing its title and abstract for its relevance to the topic. Finally, we retrieved 37 studies in line with our objective. Furthermore, 22 were excluded due to the mentioned reasons in the PRISMA flow diagram (**Fig 1**). Then, 16 studies reporting

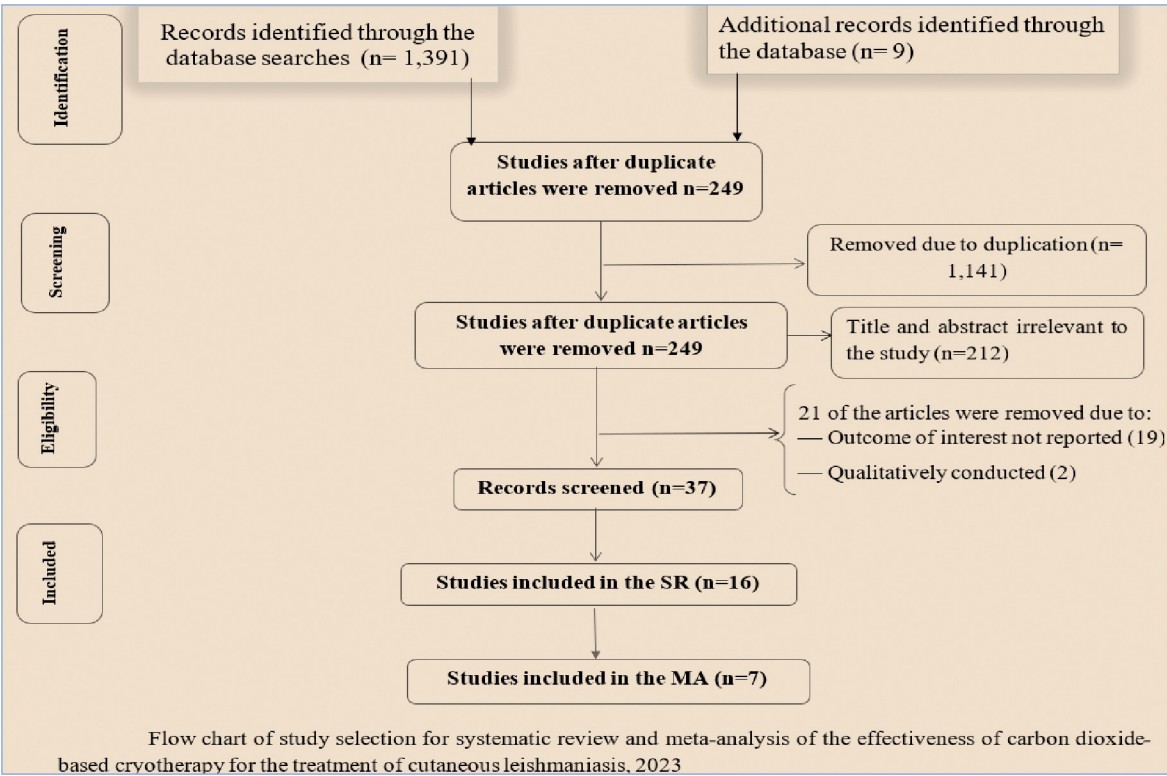

**Fig 1. Flow chart of study selection for systematic review and meta-analysis of the effectiveness of carbon dioxide-based cryotherapy for the treatment cutaneous leishmaniasis, 2023.**

treatment outcomes of carbon dioxide-based cryotherapy for the treatment of CL were included in the systematic review, of which seven studies were used for further meta-analysis.

## Characteristics of included primary studies

Most of the original research One study (6.7%) published since 1991 [26], one (6.7%) published between 2004 and 2019 [14,17,19,21–29,31], and one (6.7%) study included in the review was not yet published [20] (**S4 Table**).

Countries are divided into two categories: the Old World (Europe, Africa, and Asia) and the New World (North America, South America, and the Caribbean) [10]. As a result, just 6.25% of the included papers [31] were from the New World, whereas 93.75% of the studies [14,17–21,23,24,26–29,31] were classified as old world research.

Furthermore, three (18.75%) private clinics, five (31.25%) research centers, five (31.25%) dermatological clinics, and three (18.75%) research areas were unknown (**S4 Table**).

Four studies (31.25%) [19,21,23,25] combined SSS and histopathological tests, one study (12.5%) [32] used PCR only, one study used a combination of diagnostic tests including SSS, PCR, and fine needle aspiration [24] and five studies (31.25%) [14, 20, 21, 27, 29] used SSS examination alone to diagnose patients. Nevertheless, the diagnostic methods used in four (25%) articles [18, 22, 26, 31] investigations were poorly described (**S4 Table**).

The number of lesions included in the study ranged from 1 to 5, with specified a diameter of 12 (75%) [14,18–21,23–27,31] ranging from 2.5cm to 5cm. However, 4(25%) [19,22,26,28] of the articles did not specify the diameters of the lesions (**S4 Table**).

Furthermore, more than one-third of the included publications: 4(25%) articles [19–21,27] had lesion durations of more than 6 months, while the other fourth: 5(31.25%) [14,19,23,24,29] had durations of less than 6 months the rest 7 (43.75%) [18,21,22,25,26,28,31] of the publications did not specify the duration of the lesions (**S4 Table**).

For 43.75% of the papers [18–21, 23, 25, 29], carbon dioxide cryotherapy was applied twice; for 6.25% of the papers [24], it was applied five times; and for the remaining 25% of the papers [14,21,27,31], it was applied just once. Furthermore, four (25%) of the papers [17,22,26,28,31] did not provide the number of times cryotherapy was used in each case.

As a result, the articles were divided into groups according to the treatment cure rate, which was specified as long as the lesion healed [14,22,23,26,27], after three months [18,25,29], or not at all [20, 21, 24, 28, 31].

Ultimately, following carbon dioxide-based cryotherapy, three publications [23,24,27] provide a cure rate of less than 80%, whereas the remaining thirteen [14,18–22,25,26,28,29,31] demonstrate a cure rate of more than 80%. Lastly, the age range of the subjects was stated in the articles to be between 5 and 60.

Nearly 43.75% of CL cases treated with carbon dioxide cryotherapy had regular follow-ups every four weeks [23–29]; twice a month was observed in 37.5% of cases [14,18–22]; 6.25% of cases had follow-ups once a week [19], once every couple of months [21], and none at all [31].

## Meta-analysis and meta-regression

To show a visual summary of the data in this meta-analysis, a forest plot (**Fig 2**) was used to estimate the pooled effect size and each study's effect with their corresponding confidence intervals (CI). Of the seven papers, only one showed a publication bias (**Fig 3**).

Seven studies [17,20,21,23,24,27,29] were included in the meta-analysis to evaluate the pooled value of the effectiveness of carbon dioxide cryotherapy. As a result, the forest plot revealed that the total pooled efficiency of carbon dioxide cryotherapy was 87.84% (95% confidence interval [CI] 65.92, 109.77%). As the forest plot showed, both the $I^2$ = 54.15% and the p

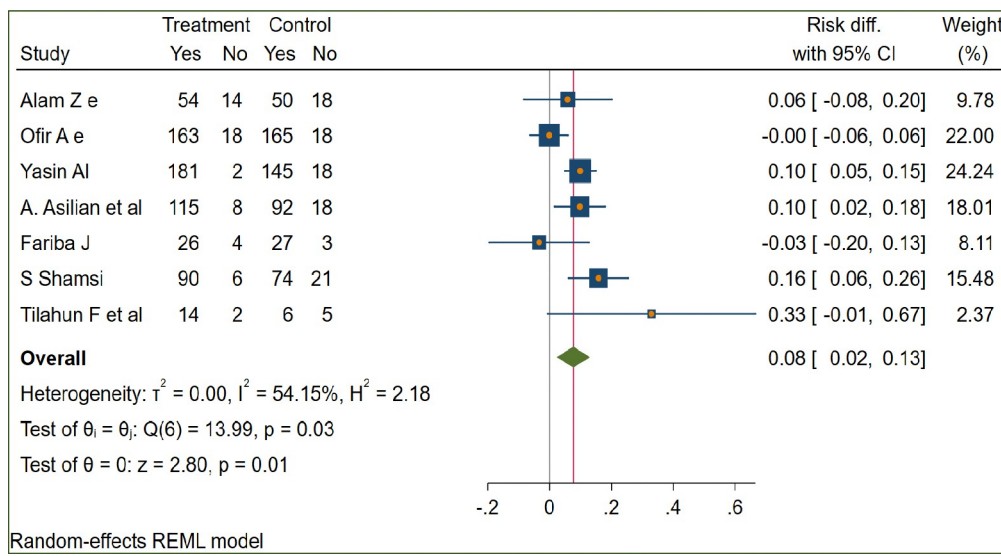

**Fig 2. Forest plot of the pooled effectiveness of carbon dioxide based cryotherapy for the treatment of cutaneous leishmaniasis, 2023.**

value = 0.03. This indicates that this systematic review and meta-analysis had substantial heterogeneity. Meaning, there is a significant variation between included primary studies. Moreover, since the treatment types for the control groups or standard treatments are variable, we are unable to pool the cure rate of the controls for the respective articles (**Fig 2**).

Subgroup analysis was performed with the help of several factors and a random effect model. The number of lesions, the size of the lesion, and repeated application of the treatment were found to be the source of heterogeneity. Subgroup analysis shows that the cure rate of cryotherapy will rise from 65.18% (95% CI: 17.16–113.20) to 94.34% (95% CI: 69.21–119.48) as the number of lesions decreases from 5 to 2 (**Figs 4**). Similarly, in the subgroup analysis of a repeated application of cryotherapy, when the treatment is applied for the second time, the cure rate was improved from 82.10% (95% CI: 50.19–114.00) (**Figs 5**) to 93.01 (95% CI: 62.72–123.29) (**Figs 6**). However, there is no statistical difference in cure rate among the primary articles (**S5 Table**).

The Galbraith plot is important to detect potential outliers. In the absence of substantial heterogeneity, we expect around 95% of the studies to lie within the 95% confidence interval (CI) region. However, one of the primary articles is out between -2 and +2; hence, there is heterogeneity (one outlier paper) (**Fig 7**), and objectively, the publication bias was checked using Egger and Begg's test (**Fig 8**).

## Meta-regression

For the effectiveness of carbon dioxide cryotherapy, the duration of the lesion 0.013 (CI: 0.0025, 0.023), the size of the lesion -0.05(-0.089, -0.011), and repeated application of the treatment had a great contribution to the cure rate of LCL (**Figs 9**).

## Sensitivity analysis

It can indicate that the effects of individual published papers influence the overall effects of meta-analysis estimates. Therefore, all papers were located within the upper and lower limits

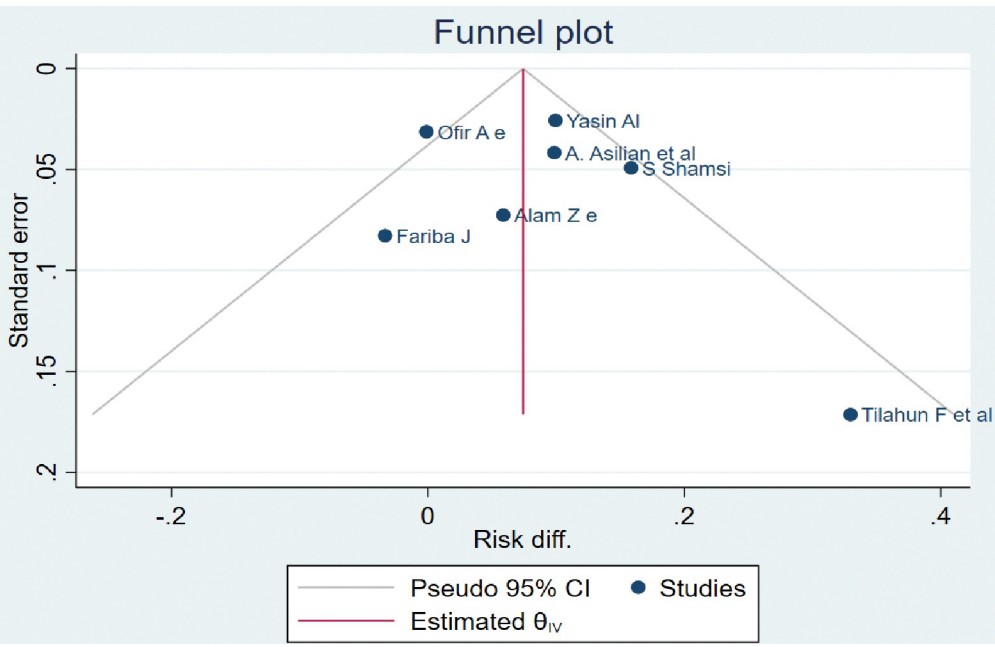

**Fig 3. Funnel plot showed the presence of publication bias estimating the effectiveness of cryotherapy among cutaneous leishmaniasis cases, 2023.**

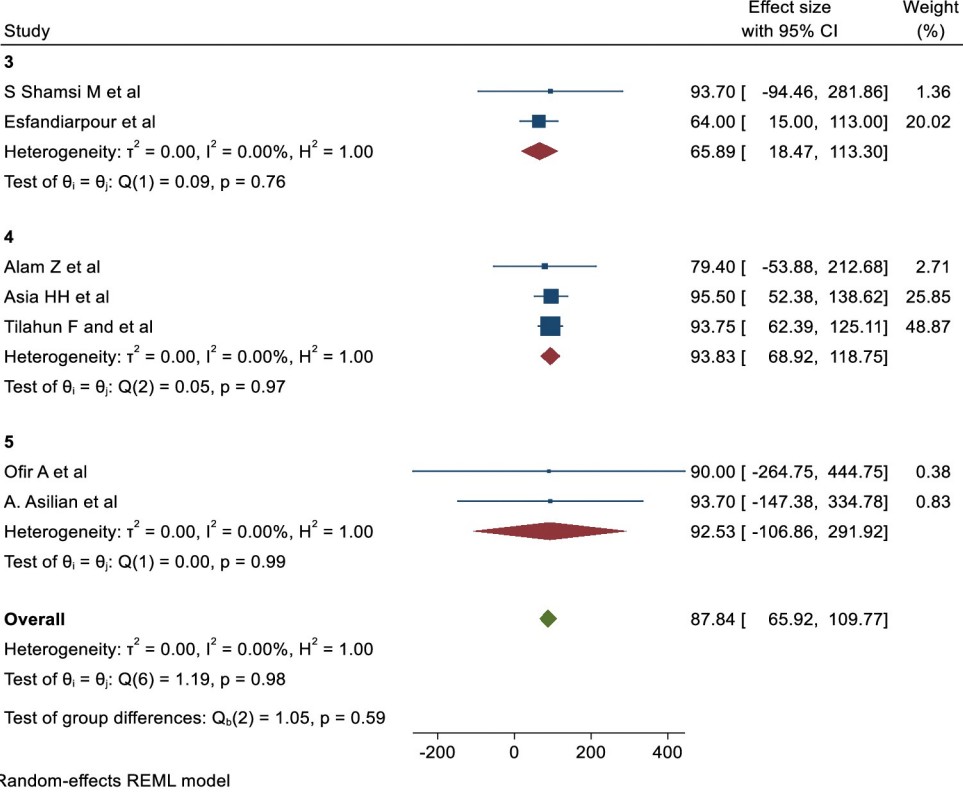

Figure 4: Sub-group analysis for the effectiveness of CO2 cryotherapy based on the size of the lesion

**Fig 4. Sub-group analysis for the effectiveness of CO2 cryotherapy on the size of the lesion.**

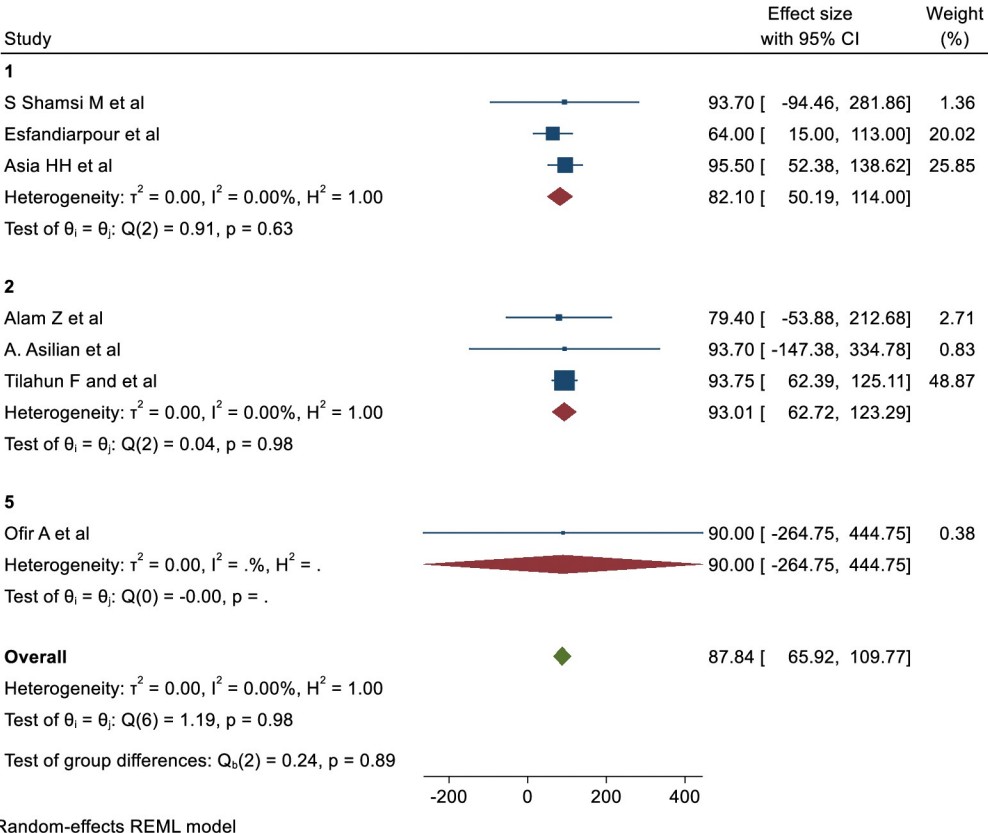

**Fig 5. Sub-group analysis for the effectiveness of CO2 cryotherapy single application of treatment.**

or CI. Hence, no single study contributes to or influences the overall effect size of the papers (**Fig 10**).

## Discussion

From this systematic review and meta-analysis, we synthesize that carbon dioxide-based cryotherapy was used in different application techniques, follow-up, and cure declared time. However, it is widely used and reported from the Old World, including Ethiopia, but rarely from the New World, i.e., only one paper was reported from the new planet. The studies reviewed showed that carbon dioxide-based cryotherapy was used every 2 weeks [18–21,31] and 4 weeks [23–29]. Moreover, the number, size, and duration of lesions to be treated with this cryotherapy were different concerning the study site, and the experience of the clinicians [22]. However, the follow-up or application period variation did not show a statistical significance in the cure rate. Such differences in application and inclusion of the cases might be the fact that unavailability of national referencing standardized treatment guidelines in the respective study sites [22]. The assessment and evaluation of the effectiveness of carbon dioxide-based cryotherapy for the treatment of cutaneous leishmaniasis would have great value for decentralizing the treatment and diagnosis plan in Ethiopia [38].

In this study, the overall pooled effectiveness (cure rate) of carbon dioxide cryotherapy for the treatment of cutaneous leishmaniasis was 87.7%. Both the treatment and controls were given for LCL with a size less than 5 and some lesions less than or equal to 4. This finding was relatively higher than the control groups. However, the control groups of antileishmanial

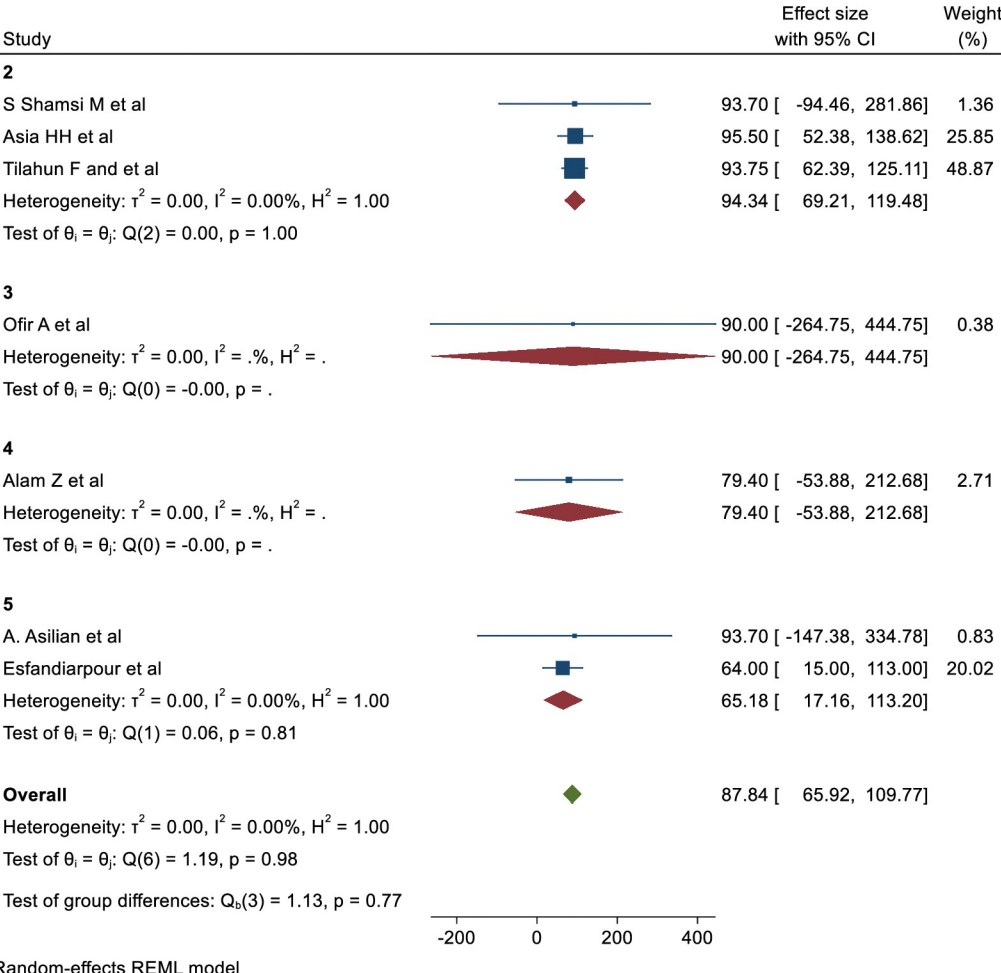

**Fig 6. Sub-group analysis for the effectiveness of CO2 cryotherapy repeated application of treatment.**

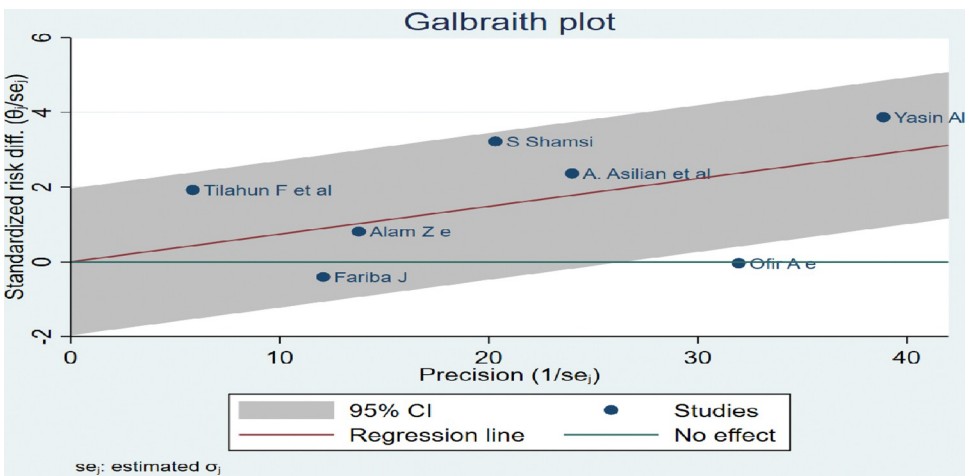

**Fig 7. Galbraith plot showed the presence of outlier studies estimating the effectiveness of cryotherapy among cutaneous leishmaniasis cases, 2023.**

```
. meta bias, egger

  Effect-size label: Risk diff.
        Effect size: _meta_es
          Std. err.: _meta_se

Regression-based Egger test for small-study effects
Random-effects model
Method: REML

H0: beta1 = 0; no small-study effects
            beta1 =       0.80
      SE of beta1 =       1.070
                z =       0.75
        Prob > |z| =      0.4550
```

**Fig 8. Egger and Begg's test showed the presence of publication bias estimating the effectiveness of cryotherapy among cutaneous leishmaniasis cases, 2023.**

agents were administered as single and combined patterns of administration. Besides, the finding was consistent with the research conducted on liquid nitrogen based on the treatment of LCL [32]. But to the contrary, the effectiveness was relatively low among LCL cases treated with intralesional sodium stibogluconate [39] and thermotherapy (TT) [15]. A review also showed that most usually carbon dioxide-based cryotherapy was used as a laser therapy for resurfacing acne and other scars, tattoos, and scar tissue due to various causes including CL-related scarring after the application of traditional agents [34]. However, it is also a common practice to treat CL lesions in the New World and cervical pre-cancerous cells in the Old World countries [40]. Furthermore, in addition to its effectiveness, carbon dioxide-based cryotherapy was easy to apply, less painful, and had minimal scar and post-inflammatory hyperpigmentation [34].

The effectiveness of the use of carbon dioxide-based cryotherapy for the treatment of CL was highly influenced by the size, and number of the lesion, and the application cycle of the treatment in most of the primary studies. However, in the subgroup analysis, as the carbon dioxide-based cryotherapy application for the treatment of CL repeated, the effectiveness of CL also increased from a single application 82% (50.19, 114.00) to the second treatment 86.41% (34.46, 138.36). For the local treatments including IL SSG [39] and liquid nitrogen [32] based cryotherapy, the effectiveness was relatively low to carbon dioxide-based cryotherapy. However, pain while administration of the treatment to the CL lesion [22,39], cost of the treatment [41] safety [38,40], and harmonious application of the treatment by the professionals was not easy with the use of other local CL treatments. However, the use of carbon dioxide-based cryotherapy results low to nil degree of post-inflammatory hyper and hypopigmentation macule, patch, and scar formation [38,41].

However, the effectiveness of cryotherapy may vary depending on the Leishmania species involved. Studies have shown that cryotherapy yields satisfactory results against species like *L.*

```
        _meta_es │ Coefficient  Std. err.      z    P>|z|     [95% conf. interval]
─────────────────┼──────────────────────────────────────────────────────────────
   Numberoflesion │   .0465342    .0239962    1.94   0.052    -.0004975    .0935658
 Sizeofthelesioncm │  -.0500778    .0198049   -2.53   0.011    -.0888948   -.0112608
Durationofthelesionmon │  .0127406   .0052422    2.43   0.015     .002466    .0230151
             _cons │   .0594257    .085413     0.70   0.487    -.1079807    .2268322

Test of residual homogeneity: Q_res = chi2(3) =  2.51    Prob > Q_res = 0.4742
```

**Fig 9. A table shows meta-regression outputs for the factors associated with the effectiveness of carbon dioxide cryotherapy for the treatment of localized cutaneous leishmaniasis, 2023.**

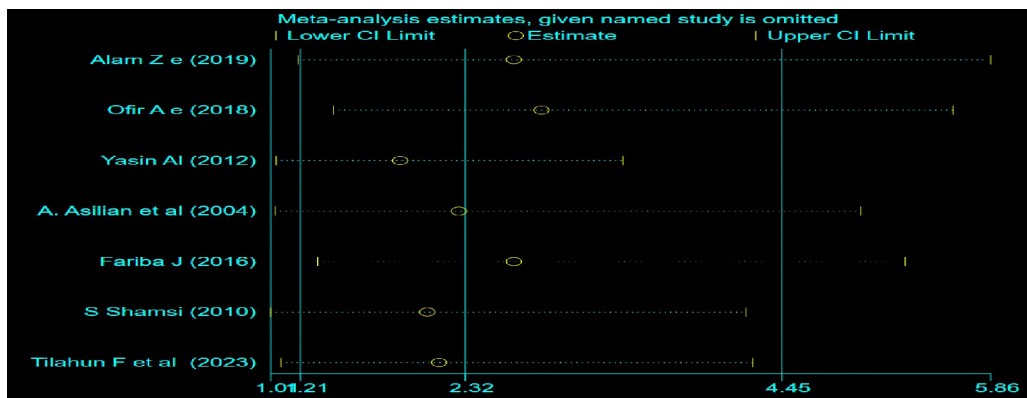

**Fig 10. Sensitivity analysis estimating the effectiveness of cryotherapy leishmaniasis cases, 2023.**

*major* and *L.donovani*, with high healing rates reported. However, some species, such as *L. brazilinesis*, may respond less favorably to cryotherapy [3,38].

## Strengths and limitations of the study

The strength of this review is its inclusion of all available shreds of evidence, encompassing both observational and interventional study designs. This comprehensive approach provides a more complete understanding of the effectiveness of carbon dioxide-based cryotherapy.

However, routine care practices for cutaneous leishmaniasis can vary significantly across countries and regions. This variability may have influenced the assessment of interventions in individual studies and the pooled estimates. This limitation raises concerns about the generalizability of the findings.

The absence of many studies from Ethiopian regions makes it challenging to extrapolate the results to the national level. This limitation highlights the need for further research in underrepresented regions.

The lack of comparable studies in the study country and worldwide makes it difficult to compare the findings with existing evidence. This limitation emphasizes the need for more research to establish a robust evidence base.

## Conclusions

This meta-analysis and systematic review provided a number of pieces of information on the effectiveness of carbon dioxide-based cryotherapy and other treatment options for cutaneous leishmaniasis. This technique is friendly, minimal adverse events, and effective. Few carbon dioxide cryotherapy studies were performed in the new world; the majority were from the old world. Success in carbon dioxide cryotherapy depends on the number of lesions being less than or equal to two, the size, the duration, and the number of applications. However, serious problems regarding study heterogeneity were found in the review. Therefore, future research aims to overcome study heterogeneity, expand regional representation, and contrast results with pre-existing evidence.

## Supporting information

**S1 Table. Search strategy for articles on the effectiveness of carbon dioxide-based cryotherapy for the treatment of cutaneous leishmaniasis, 2023.**
(DOCX)

**S2 Table. The quality of non-randomized studies using the NOS scale.**
(DOCX)

**S3 Table. The risk of bias assessment for randomized studies (RoB 2).**
(DOCX)

**S4 Table. Descriptive summary of primary studies included in the systematic review and meta-analysis of Carbon dioxide base cryotherapy for the treatment of cutaneous leishmaniasis, 2023.**
(DOCX)

**S5 Table. Sub-group analysis for the effectiveness of CO2 cryotherapy based on the size and number of lesions and number of re-treatments.**
(DOCX)

## Author Contributions

**Conceptualization:** Feleke Tilahun Zewdu, Kassahun Alemu Gelay.

**Formal analysis:** Feleke Tilahun Zewdu, Bisrat Misganaw Geremew.

**Methodology:** Feleke Tilahun Zewdu, Kassahun Alemu Gelay.

**Software:** Feleke Tilahun Zewdu.

**Supervision:** Bisrat Misganaw Geremew, Kassahun Alemu Gelay.

**Writing – original draft:** Feleke Tilahun Zewdu.

**Writing – review & editing:** Endalamaw Gadisa Belachew.

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
