## [Decision Letter · Decision Letter 0]

11 Aug 2024

Dear Mr Tilahun,

Thank you very much for submitting your manuscript "Effectiveness of carbon dioxide cryotherapy for the treatment of cutaneous leishmaniasis: systematic review and meta-analysis" for consideration at PLOS Neglected Tropical Diseases. As with all papers reviewed by the journal, your manuscript was reviewed by members of the editorial board and by several independent reviewers. In light of the reviews (below this email), we would like to invite the resubmission of a significantly-revised version that takes into account the reviewers' comments. 

We cannot make any decision about publication until we have seen the revised manuscript and your response to the reviewers' comments. Your revised manuscript is also likely to be sent to reviewers for further evaluation.

Sincerely,

Claudia Ida Brodskyn

Section Editor

Claudia Brodskyn

Section Editor

Reviewer's Responses to Questions

**Key Review Criteria Required for Acceptance?**

**Methods**

-Are the objectives of the study clearly articulated with a clear testable hypothesis stated?

-Is the study design appropriate to address the stated objectives?

-Is the population clearly described and appropriate for the hypothesis being tested?

-Is the sample size sufficient to ensure adequate power to address the hypothesis being tested?

-Were correct statistical analysis used to support conclusions?

-Are there concerns about ethical or regulatory requirements being met?

Reviewer #1: The study design is appropriate to address the stated objectives

Reviewer #2: This manuscript presents an intriguing summary of the evidence regarding the effectiveness of carbon dioxide cryotherapy in treating localized cutaneous leishmaniasis, aiming to support its continued clinical use. The authors included 16 studies, of which only 7 underwent meta-analysis, covering a total of 1,357 cases

Overall, the study adhered to the standard methodologies for this type of research and manuscript preparation. Nonetheless, there are several issues within the manuscript that require attention, as well as some that were omitted and could have varying implications for data analysis and conclusion drawing.

Rows 99 and 100 mention the following about carbon dioxide base cryotherapy: It is relatively easy to apply (17), and effective (less than 10 % of failure 100 rate) (15,18), minimal adverse events (16,19) and it takes shorter healing time (15,19). All references cited in this paragraph however are not specifically related to CL, and the reader might be misled to believe that these numbers refer to CL 

Rows 109-112 mention issues related to patients' barriers to CL treatment and that AHRI will conduct a study on this topic at a community level. I do not see how it fits in the manuscript (at least in the introduction section) 

Row 122-124: mention the use in Ethiopia of laser and the lack of well-organized evidence for the effectiveness of this laser therapy on CL caused by L. aethiopica. Personally, the mention of laser in this paragraph creates confusion and more important is not the topic of this manuscript. 

Rows 155-161: I found it quite difficult to understand the whole paragraph: It mentions that The outcome of interest for this review was cure rate (effectiveness) of cutaneous leishmaniasis after application of CO2 laser or cryotherapy. But it was mentioned before that “This study aims to summarize the evidence on the effectiveness of carbon dioxide cryotherapy for the treatment of localized cutaneous”. 

How the treatment was applied and patients follow-up after the application across the different studies is not clear: Rows 158-161 mention the following schemes: 

• once per two weeks till the lesion gets away (cure) 

• or not healed up to three months. 

• The lesions were treated 4-6 times every two weeks. 

Regarding follow-up, it only mentions the following:

When the lesion gets cured within the 4x or less, the patients were had follow up only to see the presence of any satellite lesion around the intervention sites.

The number of applications, time intervals between applications, and the exact follow-up period to assess cure after the onset of treatment are crucial pieces of information to better assess the study results and all of them are not provided.

There is a lack of information regarding the criteria or parameters used to define ‘cure’ or ‘no cure.’ Additionally, there is no data on the devices used to deliver CO2, which, based on other studies such as those on acne, is an important variable in terms of efficacy

It is well-acknowledged in Clinical Leishmaniasis that treatment outcomes are heavily dependent on the species of Leishmania causing the lesions. No single reference or analysis has been presented here that considers this.

**Results**

-Does the analysis presented match the analysis plan?

-Are the results clearly and completely presented?

-Are the figures (Tables, Images) of sufficient quality for clarity?

Reviewer #1: The analysis presented match the analysis plan.The analysis are presented match the analysis plan.The author needs to display the location, number, and comparison images of the skin lesions before and after treatment in the results.

Reviewer #2: There are several inconsistencies in the Results section.

Rows 191-193 indicate that 16 studies reporting treatment outcomes of carbon dioxide based cryotherapy for the treatment of CL were included in the systematic review, of which seven studies were used for further meta-analysis.

However in the following section, none of the numbers mentioned add up to 16. See some examples below: 

The majority of the primary studies 13(87%) were published between 2004 to 2019, one (6.7%) since 1991 (24), and one (6.7%) study included in the review were not published yet. (Total 15)

Moreover, 5 (3.33%) of the research were conducted in research centers, 4 (26.7%) in dermatology clinic, 3 (20%) private clinic and 2 (13.33%) research area was unknown. (Total 14)

In 9 (53.33%) of the cases, SSS exams were used to diagnose the patients, 5 (33.33%) of the

cases combined SSS and histopathologic exams, and 2 (15%) research used PCR; however, 4

 (30%) of the studies' diagnostic procedures were not precisely defined. (Total 20)

In other instances, only % are reported: Overall, the majority (73.3%) of the studies were conducted in old world (25) where as 26.7% of the primary 200 studies were conducted in new world countries (25, 26, 27).

Furthermore, more than one-third of the included publications (30, 31, 37) had lesion durations of more than 6 months, while the other two-thirds (24, 29, 32, 37) had durations of less than 6 months. One-third of 16 is ~5 and only 3 papers are cited, which by the way is the same number of manuscripts described in Table 3. Perhaps its easier to say 3 out of 16 publications reported lesions of >6m

Rows 221–216 mention the number of treatments applied: 2 times: 10; 5 times: 1; 1 time: 3 (Total 14)

Etc, Etc.

217-219: At what time (after onset of treatment) was cure defined? 

Rows 237-238: Similarly, when the treatment is applied a second time, the cure rate increases dramatically from 82.10% (95% CI: 50.19 - 114.00) to 93.01 (95% CI: 62.72 - 123.29). I wonder if the statement of “increases dramatically” is based on additional statistics done by the authors, I so, please add it. Looking at the 95% CI of both figures, I believe there are no statistically significant differences to make such statement.

**Conclusions**

-Are the conclusions supported by the data presented?

-Are the limitations of analysis clearly described?

-Do the authors discuss how these data can be helpful to advance our understanding of the topic under study?

-Is public health relevance addressed?

Reviewer #1: The conclusions supported by the data presented.

Reviewer #2: Discussion / Conclusions

I believe the study had several limitations, (some of which were only briefly mentioned in Rows 293-299) that drawing such categorical conclusions, as presented in the manuscript, could potentially mislead the readers.” I believe it needs to be rephrased based on the limitations found and other considerations mentioned above.

Rows 254-256: I am not sure if in the following statement “However, it is often used in the new world but infrequently in old world including Ethiopia” the authors are referring to the infrequent use of carbon dioxide-based cryotherapy for CL in particular or for dermatological use in general. For CL, there are clearly more studies in the OW than in the NW. 

Rows 267-268 briefly mention the inclusion of control groups in certain studies. It is important to elaborate on this point and compare the cure rates of the control groups with those treated with carbon dioxide-based cryotherapy.

**Editorial and Data Presentation Modifications?**

Reviewer #1: I recommend “Major Revision”.

Reviewer #2: (No Response)

**Summary and General Comments**

Reviewer #1: This is an interesting article, but some changes need to be made before publication. Firstly, the author needs to display the location, number, and comparison images of the skin lesions before and after treatment in the results. Secondly, discuss the application of this method in skin diseases in the conclusion and compare it with other treatment methods. The author needs to improve the writing and grammar of the article, for example, there is a grammar error in line 254 of the discussion. Should the syntax of "we synthesizes" be singular? Therefore, it is necessary to polish the article and provide a certificate.

Reviewer #2: (No Response)

PLOS authors have the option to publish the peer review history of their article (what does this mean?). If published, this will include your full peer review and any attached files.

Reviewer #1: Yes: Wenzhong Xiang

Reviewer #2: No
---

## [Editor Report · Decision Letter 1]

6 Sep 2024

Dear Mr Tilahun,

Thank you very much for submitting your manuscript "Effectiveness of carbon dioxide cryotherapy for the treatment of cutaneous leishmaniasis: systematic review and meta-analysis" for consideration at PLOS Neglected Tropical Diseases. As with all papers reviewed by the journal, your manuscript was reviewed by members of the editorial board and by several independent reviewers. In light of the reviews (below this email), we would like to invite the resubmission of a significantly-revised version that takes into account the reviewers' comments. 

We cannot make any decision about publication until we have seen the revised manuscript and your response to the reviewers' comments. Your revised manuscript is also likely to be sent to reviewers for further evaluation.

Sincerely,

Claudia Ida Brodskyn

Section Editor

Claudia Brodskyn

Section Editor
---

## [Decision Letter · Decision Letter 2]

12 Nov 2024

PNTD-D-24-00369R2Effectiveness of carbon dioxide cryotherapy for the treatment of cutaneous leishmaniasis: systematic review and meta-analysisPLOS Neglected Tropical Diseases Dear Dr. Tilahun, Thank you for submitting your manuscript to PLOS Neglected Tropical Diseases. After careful consideration, we feel that it has merit but does not fully meet PLOS Neglected Tropical Diseases's publication criteria as it currently stands. Therefore, we invite you to submit a revised version of the manuscript that addresses the points raised during the review process. Please submit your revised manuscript within 30 days Dec 12 2024 11:59PM. If you will need more time than this to complete your revisions, please reply to this message or contact the journal office at plosntds@plos.org. Please include the following items when submitting your revised manuscript:*
A rebuttal letter that responds to each point raised by the editor and reviewer(s). You should upload this letter as a separate file labeled 'Response to Reviewers'. This file does not need to include responses to any formatting updates and technical items listed in the 'Journal Requirements' section below.*
A marked-up copy of your manuscript that highlights changes made to the original version. You should upload this as a separate file labeled 'Revised Manuscript with Track Changes'.*
An unmarked version of your revised paper without tracked changes. You should upload this as a separate file labeled 'Manuscript'. If you would like to make changes to your financial disclosure, competing interests statement, or data availability statement, please make these updates within the submission form at the time of resubmission. Guidelines for resubmitting your figure files are available below the reviewer comments at the end of this letter. We look forward to receiving your revised manuscript. Kind regards, Claudia Ida BrodskynSection EditorPLOS Neglected Tropical Diseases Claudia BrodskynSection EditorPLOS Neglected Tropical Diseases

Shaden Kamhawi

co-Editor-in-Chief

Paul Brindley

co-Editor-in-Chief

 **Journal Requirements:** **Additional Editor Comments (if provided):****Reviewers' comments:** Reviewer's Responses to Questions

**Key Review Criteria Required for Acceptance?**

**Methods**

-Are the objectives of the study clearly articulated with a clear testable hypothesis stated?

-Is the study design appropriate to address the stated objectives?

-Is the population clearly described and appropriate for the hypothesis being tested?

-Is the sample size sufficient to ensure adequate power to address the hypothesis being tested?

-Were correct statistical analysis used to support conclusions?

-Are there concerns about ethical or regulatory requirements being met?

Reviewer #1: The study design is appropriate to address the stated objectives.

Reviewer #2: Thank you for the opportunity to re-review this interesting manuscript. The authors have kindly addressed previous comments and incorporated some changes, overall improving its quality.

However, there are some additional edits needed to further enhance the manuscript’s readability and clarity.

1. Ethiopian and WHO CL treatment guidelines: These are mentioned multiple times (Rows 86, 95, and 112). I suggest mentioning them once at the beginning of the introduction to avoid repetition.

2. Row 100-102: The statement about Carbon dioxide-based cryotherapy being easy to apply and effective (<10% failure rate) with minimal adverse effects is unclear that it is regarding its application to cervical cancer treatment. This sentence needs to be rephrased for clarity.

3. Row 118: Please provide a reference for the cure rate mentioned.

4. Outcome Measurement and Quality Assessment section (Rows 145 onward): Most of the information here seems more appropriate for the Results section. In this section, it might be sufficient to mention that the assessment considered various variables, including the number of Carbon Dioxide applications, time between applications, follow-up period, etc.

5. Rows 164-167: This sentence introduces confusion to the readers as it seems more like a conclusion. I suggest moving it to the discussion section

**Results**

-Does the analysis presented match the analysis plan?

-Are the results clearly and completely presented?

-Are the figures (Tables, Images) of sufficient quality for clarity?

Reviewer #1: The analysis presented match the analysis plan.

Reviewer #2: 6. Row 200: For the “Characteristics of included primary studies,” it would be simpler and easier to read if all these data were presented in a single table (example below)

Date of Publication

<2004 2(x%)

2004-2019 13

Unpublished 1

Region

NW 1

OW 15

Study site

Private clinic

Research center

Dermatologic clinic

Unknown

Diagnostic tests

# of lesions

etc

7. Row 275: There seems to be a confusion here. Based on the data presented above, only one study was done in NW, hence it is more frequently used in OW (not the opposite as mentioned in the text)

**Conclusions**

-Are the conclusions supported by the data presented?

-Are the limitations of analysis clearly described?

-Do the authors discuss how these data can be helpful to advance our understanding of the topic under study?

-Is public health relevance addressed?

Reviewer #1: The conclusions are supported by the data presented.

Reviewer #2: Conclusions: The authors are missing the opportunity to send a clear message. It needs to be rephrased, i.e.

• A small number of studies,

• the majority conducted in OW,

• problems of heterogenicity,

• duration and size of the lesion as well as the number of applications impact the effectiveness of carbon dioxide therapy

• Patients with up to 2 lesions receiving at least two applications are the ones benefiting the most (efficacy ~93%) with this intervention.

**Editorial and Data Presentation Modifications?**

Reviewer #1: Accept.

Reviewer #2: (No Response)

**Summary and General Comments**

Reviewer #1: (No Response)

Reviewer #2: (No Response)

PLOS authors have the option to publish the peer review history of their article (what does this mean?). If published, this will include your full peer review and any attached files.

Reviewer #1: **Yes: **Wenzhong Xiang

Reviewer #2: No

---

## [Editor Report · Decision Letter 3]

29 Nov 2024

Dear Mr Tilahun,

We are pleased to inform you that your manuscript 'Effectiveness of carbon dioxide cryotherapy for the treatment of cutaneous leishmaniasis: systematic review and meta-analysis' has been provisionally accepted for publication in PLOS Neglected Tropical Diseases.

Best regards,

Claudia Ida Brodskyn

Section Editor

Claudia Brodskyn

Section Editor

Shaden Kamhawi

co-Editor-in-Chief

Paul Brindley

co-Editor-in-Chief

---

## [Editor Report · Acceptance letter]

17 Dec 2024

Dear Mr Tilahun Zewdu,

We are delighted to inform you that your manuscript, "Effectiveness of carbon dioxide cryotherapy for the treatment of cutaneous leishmaniasis: systematic review and meta-analysis," has been formally accepted for publication in PLOS Neglected Tropical Diseases.

Best regards,

Shaden Kamhawi

co-Editor-in-Chief

Paul Brindley

co-Editor-in-Chief
